# The Influences of Supportive Leadership and Family Social Support on Female Managers’ Organizational Effectiveness: The Mediating Effect of Positive Spillover between Work and Family

**DOI:** 10.3390/bs13080639

**Published:** 2023-07-31

**Authors:** Yoonhee Park, Jiyoung Kim, Harin Lee

**Affiliations:** 1Department of International Office Administration, Ewha Womans University, Seoul 03760, Republic of Korea; yoonhpark@ewha.ac.kr; 2Hyundai Mobis Technical Center of Korea, Yongin 16891, Republic of Korea; harin.lee@aiesec.net

**Keywords:** supportive leadership, family social support, positive spillover between work and family, organizational effectiveness, female manager

## Abstract

This study aims to examine the influence of supportive leadership and family social support for female managers on organizational effectiveness and test the mediating effect of positive spillover between work and family (PSWF). This study utilized data of 974 married female managers from the 6th Korean Female Manager Panel (KWMP) survey to analyze the relationship between the latent variables. Hypotheses of this study were tested using Structural Equation Model Analysis (SEM). This study found that supportive leadership and PSWF have a positive influence on female managers’ organizational effectiveness. However, family support had no significant effect on the organizational effectiveness of female managers. The analysis showed that supportive leadership and family social support positively influenced female manager’s PSWF. Also, PSWF mediated the relationship between family social support and organizational effectiveness as well as between supportive leadership and organizational effectiveness. This study provides a better understanding of PSWF as a mediator between family social support and organizational effectiveness. Contrary to previous studies that focused on the negative effects of work-family conflicts, this study highlighted the role of PSWF, justifying the need for governmental or organizational programs to increase PSWF.

## 1. Introduction

As female-labor market participation and awareness of gender equality grow, males and females are increasingly carrying out multiple roles in both the work and family domains. According to data released by Statistics Korea, the dual income household population ratio stood at 46.3% in October 2018, and though it decreased slightly in 2019, it returned to 46.3% in 2021. This rising proportion of working couples has been driven by government work-family balance policies that have fostered work environments that help married females continue their careers. However, since female’s child rearing and/or housework responsibilities prevent them from working as much as males outside the home, the gender gap in weekly working hours has remained the same [1].

Work-family balance is very important as both work and family are important parts of a fulfilling personal life. By increasing stability and vitality at home, individuals who harmonize their work and family lives increase the quality and comfort of their family members’ lives and enjoy improved personal efficiency in the workplace. However, individuals with multiple roles in both work and family areas often struggle to maintain work-family balance. For instance, the 2018 Work and Family Balance Indicators of Statistics Korea showed that the gap in employment-population ratios between unmarried males and females stood at 1.6%, whereas the gap in employment-population ratios between married males (81.9%) and females (53.4%) widened to 28.5% points [2]. This undesirable situation in employment for married females reveals a reality that does not reflect changing perceptions of female’s roles in the division of household affairs and child rearing.

Meanwhile, previous studies show that family-friendly policies with the support of senior leaders ‘produced higher levels of job satisfaction and increased perceived organizational performance’ [3,4,5]. However, organizational members’ concerns about negative career impacts may make them reluctant to use such family-friendly systems if the organizational atmosphere or the boss’s attitude toward these systems are not favorable. As such, identifying the exact mechanism for reducing work-family conflicts is therefore crucial. To reduce work-family conflicts, individuals must presumably receive support from both work and family. Specifically, individuals should receive emotional support from their bosses in their workplaces or informal support through organizational climates in their work areas [6,7,8,9] At the same time, they should also receive family support via the division of family roles and responsibilities [10,11]. Since supportive leadership and family social support seem imperative to maintaining work-family balance, examining how they influence the work-family balance for female managers is critical. 

Supportive leadership is exhibited through individual attention and consideration of organizational members’ needs. Supervisors who practice supportive leadership understand and care for their members’ personal circumstances, helping to both minimize their work-family conflicts and maintain work-life balance [12]. Recognizing that family-friendly workplace policies are ineffective if bosses do not provide related information or allow members to use it, several studies have highlighted the importance of supportive bosses [13,14]. Moreover, the close relationship between supportive leadership and job performance [15], including creativity [16], has drawn more attention to supportive leadership that, research has shown, motivates members to perform better by providing them responsibility and the authority to work on their own.

Family social support refers to the emotional and instrumental support given by family members when needed [17]. Individuals’ family members can help them perform multiple roles in both the work and family domains by providing psychological support, expressing empathy, and demonstrating understanding [11,18]. Guest [10] also pointed to family culture, which encompasses family support and attitude, as a major factor affecting the work-family balance. Meanwhile, sharing family obligations such as child rearing and housework is an important component of instrumental support in families. Accordingly, emotional and instrumental support from family members help working mothers become more immersed in organizations and more engaged with their roles [19]. Put another way, to harmonize their work and family lives, female managers need consideration and support in their family relationships. 

As work-life balance has become an increasingly prominent social issue and extra effort has been devoted to reducing working hours to maintain a 52-h workweek system, companies have begun actively engaging in family-friendly management to improve their employees’ work-family balance [20]. Maintaining employees’ work-family balance has benefits for both employees and organizations. In addition to increasing employees’ life satisfaction, it can increase companies’ productivity by boosting employees’ work efficiency and organizational commitment, which can eventually lead to organizational success. These positive effects can be even more pronounced for females as work-family balance has been shown to have more positive effects on female entrepreneurs than on their male counterparts [21]. In this sense, how to increase female manages’ organizational effectiveness has become critical concerns for individuals and organizations. Due to these interests, several studies have been conducted. Kim and Han [22] showed that work-family conflicts negatively impact married female managers’ job satisfaction, while work-family enrichment positively impacts their job satisfaction and organizational commitment [22]. Koo et al. [19] also found that work-family balance combined with supervisory support and horizontal organizational culture increases working mothers’ organizational commitment. 

However, previous studies have investigated whether positive spillover between work and family is related to individual’s higher life satisfaction or career success rather than organizational effectiveness [23,24]. In addition, previous literature [19,22] did not uncover how supervisory and family support could lead to female managers’ higher level of organizational effectiveness. Therefore, further research that examines in what social context it would be effective to increase organizational effectiveness of female managers is required, using positive spillover between work and family based on work-family spillover theory from the perspective of human resource development. Apart from the concept of role conflict, which was mainly discussed to measure the degree of work-family compatibility, this study focused on the synergy effect within the work-family area [25]. Thus, the purpose of this study was to examine the influence of supportive leadership and family social support on the organizational effectiveness of female managers and to analyze the mediating effect of PSWF. This study tried to determine if individual’s satisfaction in the workplace affects family life and positive experiences at home affects the work life in accordance with the work-family spillover theory. In addition, a dependent variable was set as organizational effectiveness, comprising organizational commitment and job satisfaction as its sub factors, which are representative indicators of dealing with psychological aspects.

## 2. Literature Review and Hypotheses

### 2.1. Organizational Effectiveness

Organizational effectiveness has been defined in various ways according to the researcher’s subjective perspectives, research purposes, and measurement indicators. Since it contains diverse and comprehensive meanings, it is difficult to define it as one. Nevertheless, organizational effectiveness can be quantified based on how effectively organizations operate in terms of human resource development and management, and how effectively organizational goals have been achieved. Different scholars have identified different sub-factors of organizational effectiveness. For instance, Campbell [26] organized the components of organizational effectiveness into economic, administrative, and psychological indicators, and measured job satisfaction, organizational commitment, and motivation as sub-factors [27,28]. Job satisfaction and organizational commitment are useful sub-factors of organizational effectiveness [22,29,30]. Many researchers present organizational commitment and job satisfaction as indicators of organizational effectiveness because organizational effectiveness is related to the behavior and measurability of organization members [30]. Moreover, it is important for management to understand the attitudes and behaviors of employees that can affect organizational management [31].

Job satisfaction refers to the satisfaction of workers in relation to their duties; it is considered a positive and pleasant emotional condition associated with an individual’s job and job experience. It can be divided into intrinsic satisfaction, which encompasses psychological and emotional satisfaction, and extrinsic satisfaction, which encompasses the various forms of visible compensation [32]. Tiffin and McCormick [33] also defined job satisfaction as a positive status based on the results of workers’ performance. Since job satisfaction can be directly linked to organizational productivity, Lichtenstein [34] defined it as an emotional result of compensation based on the job and job experience performed by an individual. Job satisfaction’s association with job performance has added to its importance at both the individual and organizational levels. Namely, high job performance is more likely to appear for those who feel satisfaction regarding their duties because they can maintain good relationships with their organizations and increase organizational effectiveness. On the other hand, job dissatisfaction negatively affects turnover and absenteeism.

Organizational commitment refers to the extent to which members of organizations are psychologically attached to, identify with, and commit to their organizations or organizational goals [35]. Allen and Meyer [36] divided organizational commitment into the following three types: affective commitment, continuance commitment, and normative immersion. Ferris and Aranya [37] described that organizational commitment as the extent to which individuals identify with and are willing to immerse themselves in and remain committed to their organizations. Meanwhile, Eisenberger et al. [38] stated that organizational commitment refers to the passion of an organization’s members for said organization, including their relationships with others, groups, and organizations. Organizational commitment plays an important role in explaining organizational effectiveness because it represents individuals’ devotion and commitment to their organizations; this highlights the importance of managing and increasing organizational commitment [29,39].

Several studies have found positive relationships between supportive leadership, family social support, and organizational effectiveness. Several studies found that task performance was positively related to individuals’ perceptions of supportive leadership [40,41] and that team commitment had a mediating effect [42]. Similarly, significant and positive relationships were observed between both supportive culture and job satisfaction and supportive culture and organizational commitment [43]. Moreover, organizational work-family support has been shown to influence organizational commitment and job satisfaction [44]. Lastly, researchers have found leaders’ organizational identification, affective organizational commitment, job involvement, in-role behaviors, organizational citizenship behaviors, employee job satisfaction, well-being, and self-efficacy to be positively related to upper-level supportive leadership climates [45]. 

Several studies have also revealed a positive relationship between PSWF and work effectiveness. For instance, job satisfaction and organizational commitment have been found to be improved by positive work-to-family and family-to-work spillover [23,46]. Moreover, family support for employees’ work issues have been linked to their job satisfaction via work-family balance, and supervisor support for her family issues have been shown to have a positive and direct effect on job satisfaction [47].

### 2.2. Supportive Leadership

Supportive leadership is one of the four leadership types—directive, supportive, participative, and achievement-oriented—identified by the path-goal theory of leaders [48]. As a type of leadership behavior, it is characterized by the provision of emotional support for employees and expressions of concern for employees’ needs and welfare [48,49]. In addition to focusing on the personal needs and welfare of organization members, supportive leadership involves active efforts by team leaders to create favorable work environments and both meet the needs and improve the well-being of team members [50]. According to House’s [51] concept of supportive leadership, supportive leaders listen to their followers’ problems, express understanding and concern, provide feedback and task-related information, and help followers perform their work. According to the path-goal theory, such leaders motivate their followers by selecting the appropriate type of leadership (directive, supportive, participative, and achievement-oriented) for a given situation, thereby inducing positive attitudes and improved performance from them [51]. The path-goal theory has validated both directive and supportive leadership; directive leadership involves leadership actions that require members to follow rules and procedures, while supportive leadership focuses on the needs and welfare of members [48]. Both directive and supportive leadership have been shown to significantly impact team effectiveness, productivity, and learning [52]; however, research has shown that supportive leadership has a more positive impact on the attitudes and performance of the members than directed leadership [53]. Similarly, Zaman et al. [41] found that supportive leadership has positive effect on sustainable project success while Simorangkir et al. [40] verified that supportive leadership positively influences learning culture and job performance.

Empirical evidence has revealed a positive relationship between supportive leadership and organizational effectiveness [54,55,56,57,58,59,60,61,62,63]. For instance, the supportive leadership style has positively and significantly predicted employee job satisfaction [60]. Moreover, Abdullah et al. [54] proved that employees perform better when they perceive themselves to be supported by other employees and management. Smit et al. [63] study also verified that supportive leadership facilitates employee job attachment. Furthermore, Elsaied [56] found that supportive leadership has a positive and essential effect on employee advocacy. 

In addition to fostering friendly work environments and care for the general well-being of employees [62], supportive leadership influences how employees perceive their jobs; when leaders are seen to be fair and to acknowledge employees’ good performance, employees tend to be more enthusiastic in the workplace. Since supportive leaders recognize the individual interests of employees and help them reconcile those interests with organizational objectives [59], the perception among employees that their leaders are seeking to fulfill their needs may bolster the sense of obligation they feel to meet their leaders’ needs, thereby generating more productive behaviors and performance [57]. In particular, supportive leadership encourages team members to strenuously engage in strategically important tasks [58]. Moreover, supportive leadership encourages subordinates to exercise independent initiative, clarifies responsibility, and emphasizes relationships and trust [55,64]. For these reasons, supportive leadership, which focuses on building relationships and providing psychological support for followers, has been highly recommended when the tasks assigned to employees are highly stressful and frustrating [62] or repetitive and structured [61]. These findings suggest that supportive leadership plays an important role in transforming employees’ negative work commitments into positive work commitments.

### 2.3. Family Social Support

Social support from co-workers and family members helps people harmonize the multiple roles they play in both the work and family domains [65]. According to Greenhaus and Parasuraman [17], family social support refers to the emotional interest, instrumental assistance, and information provided by family members. King et al. [66] described emotional support as the affection, interest, and assistance family members provide to encourage and understand workers, and instrumental support as the tangible assistance in housework and child rearing needed to ensure a tranquil family life.

Research has demonstrated the importance of family social support in work-family relationships, highlighting its impact on work-family balance [10,67]. For instance, instrumental support from spouses has been shown to reduce females’ fatigue or pain of in double earner couples [68]. Moreover, researchers have found various forms of psychological support, including family support, empathy, and understanding, to be helpful in facilitating smooth performance of one’s work and family responsibilities [11]. Choi and Jung [69] examined work-family spillover factors for domestic female workers and found that spousal support had significant positive spillover effects from family to work. 

Drummond et al. [70] showed that by reducing work-family conflicts, supervisor and family support ultimately reduces the psychological strain and increases the job and family satisfaction of female employees. In a similar vein, Barnett et al. [71] found that while family support may not directly help decrease stress at work, it may reduce such stress indirectly by decreasing family-work conflicts. Specifically, employees who feel they have enough time to get everything done at home may have feelings of competence, reducing potential work resource depletion [71]. Similarly, Mansour and Tremblay [72] proved that both generic and specific work-family social support decreases job stress through work-family and family-work conflicts, while Pluut et al. [73] verified that social support at home significantly attenuates the effect of emotional exhaustion on work-family conflicts. Despite these findings regarding the relationship between family social support and PSWF, considering the important roles these factors play in the work-family relationships of married female managers, fully elucidating this relationship will require further research.

### 2.4. Positive Spillover between Work and Family

The expansion of female’s participation in social activities and the labor market has given rise to new perspectives on work-family relationships. From the perspective of blocking or removing negative factors that negatively affect work-family relationships, the advent of these changes has led to a change into the perspective of supporting work-family balance [74]. Given the fact that PSWF contributes to improved work and family life quality [25], further research is needed for female managers.

The interaction between work and family life is primarily explained by the theory of transition. According to transition theory, emotions, attitudes, and behaviors gained in one area shift to other areas [75]. The concept of work-family spillover encompasses both positive and negative transitions from work to family and family to work; work-family spillover occurs when emotions, behaviors, or attitudes that arise from a role in a work or family area affect the feelings, behaviors, or attitudes in the other area [25,65]. Considering the direction of transfer, Grzywacz and Marks [76] divided spillover into four categories: negative spillover from work to family, negative spillover from family to work, positive spillover from work to family, and positive spillover from family to work. Applying the concept of interference and enhancement, Wadsworth and Owens [77] classified negative transition as work interference with family and family interference with work, and positive transition as work enhancement of family and family enhancement of work. The fact that work-family balance policies have been implemented in Korea makes it necessary to conduct an integrated analysis of the work-family relationship and thereby confirm the positive interactions between work and family for the development and utilization of female workforce [74]. 

Positive experiences in the home domain may have positive spillover effects on the work domain, while negative experiences can lead to negative spillover [78]. In fact, several studies have examined the effects of PSWF. For instance, two meta-analyses have positively linked both work-to-family and family-to-work enrichment (positive spillover) to job satisfaction [79,80]. Andersz et al. [23] found that positive spillover from home to work was associated with higher life satisfaction, and the opposite was true for negative spillover. Moreover, Lim and Yu [24] validated that family-work positive spillover leads to career success of female team leaders with a partial mediating effect of leadership, highlighting the importance of family social support and positive relationships within the family domain. In the same manner, research has shown that work-family facilitation has a greater effect on managerial competence, job satisfaction, and organizational commitment than work-family conflicts [81]. Kim and Han [22] examined the work-family interface and organizational outcomes in female managers and found that reducing work-family conflicts and enhancing work-family enrichment positively contributes to job satisfaction and organizational commitment. Similarly, Sok et al. [82] showed that investing in employees, both female and male, can pay off since the employees who experience positive spillover from home to work tend to show lower turnover intentions. 

### 2.5. Supportive Leadership and Organizational Effectiveness

Over the past decades, many researchers have verified the positive effects of supportive leadership on employee satisfaction [83,84,85,86] and job stress [87,88]. Support from supervisors may decrease employees’ negative feelings about their jobs [89], exerting a buffering effect on members’ occupational stress [87]. In addition to contributing to employee satisfaction, supportive leadership has been shown to have a positive relationship with employee commitment and career certainty [85]. More specifically, Mukanzi et al. [90] demonstrated that perceived managerial support moderates the relationship between employee stress, burnout, absenteeism, and employee commitment. This finding aligns with the findings of other studies conducted in the United Arab Emirates [91], Hong Kong, and Australia [92]. Furthermore, a recent study, which analyzed studies on organizational effectiveness of married working women in the organization, highlighted the importance of managerial support in increasing their organizational commitment, career satisfaction, and retention intention [64]. 

In a similar vein, by analyzing diverse organizations in the United Arab Emirates, Yousef [91] demonstrated that organizational commitment, job satisfaction, and job performance are positively related to employees’ perceptions of their supervisors as consultative or participative leaders. Furthermore, Feierabend et al. [93] found that intangible support, such as family-supportive conversations in the workplace, reduces employees’ intentions to quit and increases their organizational commitment, since it leads employees to perceive their work environments as supportive [93,94]. Additionally, the organizational commitment of employees has been shown to increase when other employees share their perceptions of managerial support [94].

### 2.6. Family Social Support and Organizational Effectiveness

Previous studies have found that family social support is positively related to career development, career success, and satisfaction at work [25,95,96,97,98]. Research has shown that family support can reduce work-family stress, buffer work-family conflicts, and support overall well-being in ways that might even help individuals overcome career-related obstacles and be more satisfied [96]. Moreover, another study validated that employees with more family support achieve greater subjective career success [99]. The positive effect of family social support on workers has been repeatedly highlighted in the pandemic period. Luu [100] demonstrated the positive effect of family social support on positive stress mindset and post-traumatic growth of workers in the tourism industry, which has been severely damaged by COVID-19. However, employees with family duties might be stressed by work-family conflicts and dissatisfied with an undesirable balance between their work and family needs [101]. 

Although several studies have verified the relationship between family social support and job satisfaction [95,96,97,98], the relationship between family social support and organizational commitment has received scant attention. However, research has shown that job satisfaction predicts organizational commitment [102,103,104] and that a positive relationship exists between job satisfaction and organizational commitment [105]; thus, presumably, family social support is positively associated with both job satisfaction and organizational commitment.

### 2.7. Mediating Role of Positive Spillover between Supportive Leadership and Organizational Effectiveness

Research has shown that employees are more likely to experience PSWF when they are socially supported by supervisors and coworkers [77]. Frone et al. [97] revealed work-to-family conflict has an indirect influence on family-to-work conflict via work distress and work overload. Similarly, researchers have found a negative relationship between supportive supervisors and subordinates’ work-family conflicts, stress, and intentions to quit, and a positive relationship between supportive supervisors and positive spillover and job satisfaction [106]. This result indicates that supportive leadership can increase employees’ job autonomy, eventually leading to positive spillover from work to family. Another study suggested that while employees tend to be concerned that their family obligations will give their supervisors negative impressions, supportive supervisors alleviate such concerns [107]. Family-supportive supervisors can relieve work-related concerns that might hinder employees’ fulfillment of family duties [75]. 

Although the relationship between positive family-work spillover and organizational effectiveness has received little attention, several studies examining the impacts of taking on multiple roles in the work and family domains have demonstrated its beneficial effects [108,109]. Since negative spillover is also referred to as conflicts, with few differences in their measures [11,110,111], it might be possible to predict the relationship between positive-spillover and organizational effectiveness by analyzing the relationship between conflicts and organizational effectiveness. In fact, research has shown that work-to-family conflicts are less likely to be reported, and that job satisfaction and affective commitment tend to increase in workplaces with family supportive policies [13,112,113,114,115].

### 2.8. Mediating Role of Positive Spillover between Family Social Support and Organizational Effectiveness

Imbalanced work and family roles can decrease employee commitment by creating conflicts in organizations, which can also lead to work-family conflicts [116]. On the other hand, the psychological resources, which involve positive emotions, motivation, and energy, required to deal with tasks actively can be accumulated in the family domain [117,118,119]. Since these psychological resources can be transferred to the work domain [120], experiences in the family domain that boost employees’ psychological resources could enhance employees’ work activities [117,121,122]. In fact, Lee and Seo [123] found that support from the spouse for work had a positive influence on work-family transition of married working women, emphasizing the supportive role of the spouse. Moreover, Haar and Bardoel [124] demonstrated that positive family-work spillover is positively related to family satisfaction and negatively related to distress [124]. Lapierre and Allen [9] also found that instrumental and emotional support from family members helps employees avoid family interference with work and showed that emotional support is positively related to the employee’s physical wellbeing. 

Based on the literature discussed above, the following hypotheses are supposed:

**Hypothesis 1 (H1).** 
*Supportive leadership for female managers positively influences organizational effectiveness.*


**Hypothesis 2 (H2).** *Family social support for female managers positively influences organizational effectiveness*.

**Hypothesis 3 (H3).** 
*Positive spillover between work and family positively influences organizational effectiveness.*


**Hypothesis 4 (H4).** 
*Supportive leadership positively influences positive spillover between work and family for female managers.*


**Hypothesis 5 (H5).** 
*Family social support positively influences positive spillover between work and family for female managers.*


**Hypothesis 6 (H6).** 
*Positive spillover between work and family mediates the relationship between supportive leadership and organizational effectiveness.*


**Hypothesis 7 (H7).** 
*Positive spillover between work and family mediates the relationship between family social support and organizational effectiveness.*


Figure 1 presents this study’s conceptual framework. Specifically, PSWF is expected to mediate the relationship between supportive leadership and organizational effectiveness and the relationship between family social support and organizational effectiveness for female managers.

## 3. Method

### 3.1. Data and Sample

This study utilized data from the 6th Korean Female Manager Panel (KWMP) survey, which was conducted in 2016 by the Korean Female’s Development Institute (KWDI). The KWMP survey is the only domestic and foreign survey that tracks managers belonging to companies; a biennial survey was conducted on female managers and personnel managers of companies with more than 100 employees from 2007 to 2018 [125]. The KWMP survey investigates the personal situations, work-family balance, and corporate-organizational status of the female workforce, aiming to accumulate data that will help females continue working without interrupting their careers and move up to senior management positions. In addition, the KWMP survey also conducts surveys on companies to which female managers belong in order to better understand the current status of human resource management. As a result of the 6th KWMP survey, 1762 samples out of 3279 valid samples completed the survey, excluding situations such as maternity leave, parental leave, leave of absence, overseas secondment, and resignation. In this study, 974 married female managers out of the 1384 people who continued to maintain their jobs from the time of the 5th KWMP survey were studied. About 84.7% of all respondents were between the ages of 35 and 50 years (under 35 years: 5.4%; 50 years and older: 11.9%). About 58.7% of them had graduated from university and graduate school (university: 37.2%; graduate master’s: 18.9%; graduate doctorate: 2.6%). Approximately 51.6% of respondents were employed in midsize companies with more than 1000 employees. The respondents’ positions were evenly distributed: 23.4% were assistant managers, 31% were department managers, 29.9% were deputy general managers, and 15.8% were team managers and above. At least 99% of them were full-time employees.

### 3.2. Measures

This study used the KWMP questionnaire, which the KWDI administered, to analyze the influence of supportive leadership and family social support on organizational effectiveness for female managers as well as the mediating effect of PSWF. All questions for these four research variables were assessed using a 5-point Likert scale: supportive leadership (three questions), family social support (three questions), organizational effectiveness (twelve questions), and PSWF (six questions).

Supportive leadership was defined as leadership actions that try to create a favorable atmosphere within workgroups and take notice of the personal needs and well-being of members. Since many previous studies have used the inventory developed by Rafferty and Griffin [49] to measure supportive leadership [126,127], this study also compared and selected the questions in the KWMP questionnaire, with the measures used in Rafferty and Griffin [49]. 

Family social support was defined as sharing the burden of family care or household chores and providing emotional support. King et al. [66] developed a Family Support Inventory (FSI) for workers that included both emotional and instrumental support. Emotional sustenance encompasses encouragement, understanding, and guidance in solving problems, while instrumental assistance involves relieving home-related duties or responsibilities. Since King et al. [66]’s FSI was used in the previous studies [128], this study also compared and the selected panel items with the items introduced by King et al. [66]. 

In this study, organizational effectiveness was divided into two components: job satisfaction and organizational commitment. Job satisfaction was defined as an affirmative and enjoyable association with an individual’s job and job experience. Job satisfaction measurements were mainly based on the Minnesota Satisfaction Questionnaire (MSQ), developed by the University of Minnesota Vocational Psychology Research department [129]. The MSQ measures job satisfaction by dividing it into intrinsic, external, and general factors, and has also been frequently utilized in previous studies [130]. 

Organizational commitment was defined as the desire to believe in the purpose and values of an organization, to strive for the organization, and to maintain qualifications as an organization member. In measuring organizational commitment, this study referred to the questions developed by Allen and Meyer [36], which were used in the previous studies [81,128]. 

Finally, Positive Spillover between Work and Family (PSWF) refers to a state in which participating in different roles in work and family areas gives individuals access to diverse experiences and resources, improving their performance. Since the questions of the National Survey of Midlife Development in the United States (MIDUS) were used in previous studies [74,131], this study also utilized the MIDUS that the John D. and Catherine T. MacArthur Foundation Research Network on Successful Midlife Development collected in 1995 [76]. PSWF was analyzed in the light of the factor structure of sixteen different items—four for each dimension—that were new to the MIDUS survey [76]. 

Table 1 shows the item components of measures, Cronbach’s alpha index, and McDonald’s ω index by variable. Cronbach’s alpha for positive spillover from work to family was 0.58; this was relatively lower than other variables. However, some studies [132,133] suggest that Cronbach’s alpha criteria is 0.60 or higher, while other studies, such as Hinton [134], set Cronbach’s alpha criteria to 0.50 or higher. Moreover, McDonald’s ω for positive spillover from work to family was 0.66, which was considered as appropriate by Bagozzi and Yi [135]. Therefore, all the variables were determined to be acceptable for reliability.

### 3.3. Data Analysis

In this study, data were analyzed using the SPSS 22.0 and Amos 22.0 statistical programs. Structural equation modeling was used to test the hypotheses. The specific data analysis process is as follows. First, a frequency analysis was performed to analyze descriptive statistics, focusing on the demographic variables and general characteristics of the female managers. Second, a correlation analysis was performed to elucidate the relationships between the potential variables. Third, the measurement model was examined using confirmatory factor analysis to check how well each variable measurement explained the factors, before the structural model was analyzed. The confirmatory factor analysis covered all this study’s latent variables: supportive leadership, family social support, PSWF, and organizational effectiveness. Finally, using structural equation modeling, this study checked the fitness of the research model and examined the relationship between the potential variables, including supportive leadership, and family social support as extraneous variables and organizational effectiveness as endogenous variables. Lastly, a boot-strapping analysis was performed to examine the mediating effect of PSWF. 

The fit indices used to estimate the model’s fit were the chi-square (x^2^) statistic, goodness of fit index (GFI), Tucker–Lewis index (TLI), comparative fit index (CFI), standardized root mean square residual (SRMR), and root mean square error of approximation (RMSEA). Values of GFI, TLI, and CFI greater than 0.90 are considered as a good fit [132,136], whereas values of SRMR and RMSEA below 0.08 are determined to be an acceptable fit [137].

## 4. Results

### 4.1. Descriptive Statistics, Normality, Correlations, and Multicollinearity

This study’s research model consisted of the following six factors: supportive leadership, family social support, positive spillover work to family, positive spillover family to work, job satisfaction, and organizational commitment. As Table 2 shows, analysis of the correlations between the latent variables showed significant and positive relationships between all the variables, identifying its consistency with the direction of the research. Moreover, since problems with multicollinearity can emerge when a correlation coefficient is bigger than 0.80 [138], this study checked all the correlation coefficients and found none of them to be problematic (all between 0.126 and 0.553). If the absolute value between the displayed correlation coefficients is less than or equal to 0.2, it can be considered to have no correlation or a minor-level correlation. While 0.4 shows a weak correlation, 0.6 or more can be judged as a strong correlation [139]. The tolerance value of all variables was 0.653~0.948, and the variance inflation factor (VIF) value was 1.055~1.532. The tolerance value was more than 0.1 and the VIF value was less than 10. Thus, it can be concluded that the model is free from multicollinearity [138]. Lastly, because self-reported survey analysis can generate common method bias, this study checked all measurement items for common method bias using Harman’s one-factor test—the approach generally taken by previous studies [140]—and principal-components-analysis. The results revealed no common method bias; the total explanatory value of the first item was 30.942%, meaning it could not explain the total variance of a single item.

Minimum, maximum, mean, standard deviation, skewness, and kurtosis were measured for all the measurement variables (supportive leadership, family social support, PSWF, and work effectiveness of female managers). This study estimated multivariate normality by examining the distribution of univariate variables; the univariate normality assumption is supported when the absolute value of the skew is smaller than two and the absolute value of the kurtosis is smaller than seven [141]. In this study, all the measurement variables had standard deviations of less than three, skewness absolute values of less than two, and kurtosis absolute values less than seven; therefore, each measurement variable was predicted to have a normal distribution.

### 4.2. Exploratory Factor Analysis

Exploratory factor analysis checks whether variables measuring the same concepts are bound to the same factors. This study conducted the exploratory factor analysis using principal component analysis as an extraction method and Varimax as a rotation method, inputting all variables. All the factor loadings were above 0.4, meaning that all the variables are significant. Likewise, all the Kaiser–Meyer–Olkin (KMO) values were above 0.6, meaning the correlations between the variables are explained by other variables. Cronbach’s coefficient should be 0.6 or higher in general, and the cumulative variance should be 60% or more. 

An exploratory factor analysis was conducted for each variable; the KMO value of supportive leadership and family support was 0.658. The explanatory values of supportive leadership and family support were 40.473% and 29.864%, respectively; their cumulative variance was 70.337%, proving the validity of these factors. The KMO value of PSWF, a mediating factor in this study, was 0.803. As a result of the 6-item factor analysis, two factors were extracted in total. The first factor was related to PSWF with an explanatory power of 51.251%, and the second factor was related to PSWF with an explanatory power of 15.239%; their cumulative variance was 66.489%, verifying their validity. Finally, the KMO value of organizational effectiveness, a dependent variable, was 0.904, and two factors were extracted from the 12-item factor analysis. The first factor was related to job satisfaction with an explanatory power of 46.859%, and the second factor was related to organizational commitment with an explanatory power of 13.495%; their cumulative variance was 60.354%, proving their validity. In addition to this exploratory factor analysis, FACTOR (v. 12.04.01, 2023) program was used to confirm the results, using polychoric correlations and DWLS as an extraction method, which are considered appropriate to conduct with for Likert-type questionnaires [142]. The analysis showed the acceptable results with RMSEA = 0.079, CFI = 0.985, GFI = 1.00, TLI = 0.966.

### 4.3. Measurement Model Analysis

PSWF and work effectiveness are second-order factor models that consist of second-order latent variables. After conducting a second-order confirmatory factor analysis, this study performed a first-order confirmatory factor analysis. The results of the second-order confirmatory factor analysis satisfied the criteria for the second-order factor model goodness of fit index for both PSWF (χ^2^(p) = 37.324(0.000), χ^2^/DF(Q value) = 4.666, GFI = 0.987, TLI = 0.972, CFI = 0.985, SRMR = 0.031, RMSEA = 0.061) and work effectiveness (χ^2^(p) = 503.928(0.000), χ^2^/DF(Q value) = 9.508, GFI = 0.919, TLI = 0.902, CFI = 0.921, SRMR = 0.048, RMSEA = 0.094). After conducting a second-order confirmatory factor analysis, it is necessary to analyze construct reliability; all the PSWF items had loadings higher than 0.500, AVEs near 0.500, and Composite Reliabilities (CRs) higher than 0.700. Therefore, the convergent validity met the factor loading criteria, verifying convergent validity, AVE, and construct reliability. 

Based on the results of the second-order confirmatory factor analysis, first-order confirmatory factor analysis was conducted after converting the first-order factors into observed variables. The analysis results showed that the fit index (χ^2^(p) = 177.556(0.000), χ^2^/df(Q value) = 6.123, GFI = 0.965, TLI = 0.932, CFI = 0.956, SRMR = 0.046, RMSEA = 0.073) of the measurement model met the criteria. It was not necessary to revise the first measurement model because there were no individual measurement variables with standard loading values higher than 0.500 and multiple correlations (SMC) below 0.300.

Since the model fit and construct reliability met their respective criteria, PSWF and organizational effectiveness were both measured using two latent variables: positive spillover from work to family and positive spillover from family to work for the former, and job satisfaction and organizational commitment for the latter. Therefore, the measurement items that constitute each latent variable, the first concepts, were converted into measurement variables using item parceling. In this study, PSWF and organizational effectiveness both had two first-order factors and one second-order factor; the first-order factors were positive spillover from work to family and positive spillover from family to work for the former, and organizational commitment and job satisfaction for the latter. The parameter estimation for the initial measurement model showed that all estimates were more than 0.500. Although the AVE for family support was 0.450, which is lower than the standard value (0.500), the AVEs for the other individual measurement variables were more than 0.500, and the CRs for all the variables were more than 0.700. Thus, all the requirements for the factor loading were satisfied, verifying convergent validity, AVE, and construct reliability.

In addition, discriminant validity is related to how much a scale measuring latent variables measures other variables. Validity is high when the correlation between the variables is low. Discriminant validity is generally assured if the AVE between the two latent variables is bigger than the determinant coefficient (r^2^), the square of the correlation coefficients of each latent variable. To check the discriminant validity, the correlations between the latent variables of the measurement model and the AVE are summarized as follows. The squares of the correlation coefficients between the concepts were between 0.023 and 0.337, establishing discriminant validity, as r^2^ was smaller than AVE. Therefore, the first measurement model was selected as the final measurement model.

### 4.4. Structural Equation Model Analysis

SEM was conducted to analyze the fitness of the research model and the relationship between the latent variables to test the hypotheses of this study. The analysis results showed that the fitness index (χ^2^(p) = 177.556(0.000), χ^2^/df(Q value) = 6.123, GFI = 0.965, TLI = 0.932, CFI = 0.956, SRMR = 0.046, RMSEA = 0.073) of the structural model met the criteria; however, the direct effect of family support on organizational effectiveness was not significant. Then, the insignificant path (family social support → organizational effectiveness) was removed from the model. In this study, a chi-square test was conducted using the maximum likelihood method. As a result of the test, the modified research model became statistically significant as the degree of freedom decreased by 0.133, and χ^2^ increased by 2.147 when compared to the initial research model; thus, the modified model was selected as the final research model. 

The results of the analysis of the model fit index (χ^2^(p) = 179.703(0.000), χ^2^/df (Q value) = 5.990, GFI = 0.964, TLI = 0.934, CFI = 0.956, SRMR = 0.046, RMSEA = 0.072) showed that assessing the fitness as satisfactory is difficult because χ^2^ did not meet the criteria and the Q-value was also bigger than three; however, all the other fit indexes met the criteria, as shown in Table 3. Thus, this study concluded that the modified structural model was suitable for predicting the causal relationship between variables.

### 4.5. Hypotheses Testing

As Figure 2 illustrates, we found positive and significant relationships between supportive leadership and organizational effectiveness (0.459, *p* < 0.001), but the path between family social support and organizational effectiveness was removed from the initial model since it was insignificant. Therefore, Hypothesis 1 was supported, whereas Hypothesis 2 was not. Regarding Hypothesis 3, the effect of PSWF on organizational effectiveness was significant (0.436, *p* < 0.001). As such, the result supported Hypothesis 3. The research findings also demonstrated that the direct effects of both supportive leadership and family social support on PSWF were positively and statistically significant by showing (0.186, *p* < 0.001) and (0.575, *p* < 0.001), respectively. Therefore, Hypotheses 4 and 5 were confirmed. It was notable that the magnitude of the path between family social support and PSWF was greater than the magnitude of the path between supportive leadership and PSWF. Additionally, the squared multiple correlation coefficient (R^2^) was investigated to find the explanatory power of the endogenous variable; R^2^ of PSWF was 0.397, and R^2^ of organizational effectiveness was 0.510.

To test mediating effects (Hypotheses 6 and 7), we analyzed the direct, indirect, and total effects. Table 4 shows the standardized coefficients. The direct effect of supportive leadership on organizational effectiveness was 0.459, while the indirect effect of supportive leadership on organizational effectiveness was 0.081. Both were statistically significant at the 0.01 level. Meanwhile, the indirect effect of family support on organizational effectiveness was 0.250, statistically significant at the 0.01 level. 

Additionally, the statistical significance of the mediating effect between the latent variables was tested using the bootstrapping approach proposed by Hayes (2013), as shown in Table 5. The number of samples extracted via bootstrapping was 10,000, the indirect-effect coefficient of supportive leadership on organizational effectiveness was 0.081 (0.186 × 0.436), and the indirect effect coefficient of family support on organizational effectiveness was 0.250 (0.575 × 0.436). The fact that the upper and lower confidence limits for the indirect effect did not include 0 in the 99% confidence interval (0.167~0.634) means that the mediating effects of PSWF on the relationships between supportive leadership and organizational effectiveness and between family social support and organizational effectiveness were statistically significant. Therefore, Hypotheses 6 and 7 were supported.

## 5. Discussion

This study examined the influence of supportive leadership and family social support on organizational effectiveness through PSWF. Its findings are as follows.

First, supportive leadership had a positive effect on the organizational effectiveness of female managers. This highlights the importance of the emotional and instrumental support provided by bosses, indicating that these forms of support directly influence the job satisfaction and organizational commitment of female managers. This result is also consistent with the results of previous studies showing that supportive leadership focuses on individual support of members and has positive effects on members’ satisfaction and commitment of the members [49,54,60,85,143]. Since behaviors related to supportive leadership increase the positive effects of high-performance work practices on job satisfaction and employee engagement [144], it is worth noting the role of supportive leadership for female managers.

Second, family social support did not have a direct effect on the organizational effectiveness of female managers. This finding is consistent with the literature, where evidence of a direct relationship between family social support and organizational commitment has been somewhat inconclusive, although studies have shown that family social support has a buffering effect on managing work-family conflicts [25,96,97,98]. Demonstrating the full mediation effect of PSWF on the relationship between family social support and organizational effectiveness, this study’s results support previous studies showing that family social support reduces job stress or has a positive effect on overall job satisfaction by decreasing work-family conflicts [70,71,72]. It is, therefore, important to implement family-friendly organizational policies for employees to manage work and family responsibilities. Perceived trustworthiness and open communication within the organization result in higher job satisfaction and is positively related to policy use [145].

Third, PSWF of female managers had a significant positive effect on organizational effectiveness. As the female workforce has expanded, the relationship between work and family has been highlighted, and PSWF has come to be regarded as a key contributor to improvements in work and family quality as well as for happiness [23,25,78,82]. Consistent with previous studies showing that higher PSWF leads to higher work effectiveness [74], these results highlight the necessity of continuously checking for positive interactions between work and family for the development of the female workforce. 

Fourth, supportive leadership had a positive effect on PSWF. This result supports the findings of past research that shows that employees experience more positive ripple effects between work and family when they receive social support from their bosses and colleagues [77]. Moreover, this result suggests that individual-focused assistance from bosses and organizations can help female managers maintain and improve balanced relationships between work and family and increase happiness. Uddin et al. [146] provided important insights that it is essential for professionals and policymakers to implement family-friendly policies and facilities for the enhancement of the work–life balance of the employees in order to strengthen the engagement of the employees in the organization.

Fifth, family social support had a positive effect on PSWF. This means that increases in instrumental support, which can reduce fatigue and pain, and psychological support (e.g., family support, empathy, and understanding) can lead to increased satisfaction as well as more positive perceptions of the transition between work and family for the female managers. The results of this study align with those of previous studies identifying family support as an important contributor to work-family balance [10,68,70,72].

Sixth, PSWF mediated the relationship between supportive leadership and organizational effectiveness. The analysis of the relationship between supportive leadership and organizational effectiveness revealed both direct and indirect effects through PSWF. Specifically, supportive leadership had a significant effect on organizational effectiveness [54,63,143] and PSWF [59,62], and PSWF also had a significant effect on organizational effectiveness (*p* < 0.05). These results are consistent with those of previous studies [22,79,80,81], highlighting the importance of supportive leadership from bosses, who exercise direct influence in the workplace, in facilitating the PSWF and organizational effectiveness of female managers. On an individual level, however, female managers should realize that their positive effects in the family constitute a resource for improving their performance [147].

Lastly, PSWF mediated the relationship between family social support and organizational effectiveness. Although family social support did not directly affect organizational effectiveness, PSWF significantly mediated organizational effectiveness (i.e., job satisfaction and organizational commitment). This study analyzed the causal relationship from the perspective of PSWF. This means that family social support at a personal level can only have indirect effects on organizational effectiveness through PSWF. As such, the implementation of a program or system at an organizational or national level to help female managers recognize PSWF may increase their organizational effectiveness through PSWF.

## 6. Theoretical and Practical Implications

The academic implications of this study are as follows. First, this study contributes to the work-family positive spillover research area by revealing how supervisory and family support affects female managers’ organizational effectiveness in a Korean context. This study verified the positive influences of supportive leadership, family social support, and PSWF on organizational effectiveness. Previous studies have analyzed whether the work-family relationship directly affects management ability, job satisfaction, organizational commitment, life satisfaction, and organizational performance [22,81]; however, this study comprehensively examined the causal relationships between these three variables for female managers and verified the role of PSWF as a mediator through structural equation model analysis. Previous research had not determined whether family social support has a direct effect on organizational effectiveness. However, this study demonstrated its positive effect on organizational effectiveness through PSWF, highlighting the importance of the mediator.

Second, the fact that this study examined whether supportive leadership and family social support for female managers ultimately influences organizational effectiveness through PSWF from the perspective of positive spillover is meaningful. Previous studies of the impact of work-family relations have actively discussed the negative effects of work-family conflicts on the individual level, but the role of PSWF on individuals and organizations has only recently started to receive attention. This study verified that supportive leadership and family social support for female managers have a significant effect on organizational effectiveness through PSWF, a finding that not only underscores the role of PSWF but also justifies the need for a system or program to increase awareness of PSWF.

Therefore, the biggest contribution of the study’s findings is that PSWF is the most important factor that can improve organizational effectiveness for female managers. Given that the study findings show that work-family positive spillover plays an essential role as a mediator between a boss’s high supportive leadership and organizational effectiveness as well as family social support and organizational effectiveness, the facilitation of positive spillover between work and family is crucial to help female managers contribute to organizational effectiveness. The results of this study confirm the work-family spillover theory that the more female managers perceive themselves as receiving supportive leadership and family social support, the more significant the impact these factors have on positive spillover between work and family. This would not be possible solely by individuals or families; it is necessary to recognize the importance of female’s PSWF at both the individual and organizational levels. Furthermore, supporting and maintaining a system as well as policy for work-family compatibility at the national level is crucial. 

The practical implications of this study are as follows. First, family social support is essential to increasing PSWF among female workers. Expanding family members’ understanding and support will require the establishment and implementation of policies that develop and distribute educational materials related to increasing family social support for females’ economic activities. Second, this study confirmed that supportive leadership—bosses being friendly toward female managers and helping them to maintain balance between work and family—plays an important role in increasing the organizational effectiveness of female managers. Thus, encouraging the continuous social activities of competent female workers will require the development of various programs that expand awareness of its importance on an organizational level. Moreover, organizations should seek to foster internal cultures that enable female workers to receive assistance from bosses through supportive leadership—organizational cultures that respect and understand the lives of female managers and prioritize family-friendly policies. Third, developing relevant policies that facilitate the development of the female workforce at the national level is crucial. In addition to improving the competitiveness of companies, the economic activities of female managers play important roles in both boosting the economic status of female and promoting gender equality. Therefore, supporting career development activities for female and promoting PSWF by establishing and implementing family-friendly policies is critically important. 

## 7. Limitations and Directions for Future Research

This study empirically analyzed the mediating effect of PSWF on the relationships between supportive leadership and family support and the organizational effectiveness of female managers, and discussed the implications its findings; however, it has several limitations. First, since the analysis was conducted based on data from the 6th survey of the Korean Female Managers Panel Survey, the nature of the data itself limited the sample as it did not include female in the early stages of their careers. Therefore, follow-up studies should be conducted with a wider range of subjects as female’s career patterns vary depending on their career stages. In addition, after the COVID-19, many companies are planning to have more women at the executive level in a better work-life balance system. If the latest KWMP data reflecting this situation are used for future studies, meaningful results can be derived.

Second, an analysis of the differences between male and female managers in the relationships between supportive leadership, family social support, PSWF, and organizational effectiveness remains necessary. The concept of PSWF is mainly used to explain the work-family relationships of married female, but it also occurs for male managers. In the KWMP surveys that were conducted since 2020, male managers have also become the panels, supplementing the existing limitations. In the future, researchers should undertake comparative analyses of male and female managers and in-depth discussions of the differences between them to determine how PSWF theory is applied in real life.

Third, organizational commitment and job satisfaction were used as variables of organizational effectiveness. In addition to the psychological indicators that explain organizational effectiveness economic and administrative indicators that predict individual’s job performance need to be utilized in the follow-up studies. Using various job performance indicators at different levels will lead to a more systematic understanding of the impact of supportive leadership, family social support, and positive spillover between work and family on organizational effectiveness.

Finally, further research on various cultures and non-female manager is needed since this study was conducted on female managers in South Korea. In addition, it is necessary to contribute to the expansion of the research area by conducting comparative research between countries on positive spillover between work and family issues. 

## Figures and Tables

**Figure 1 behavsci-13-00639-f001:**
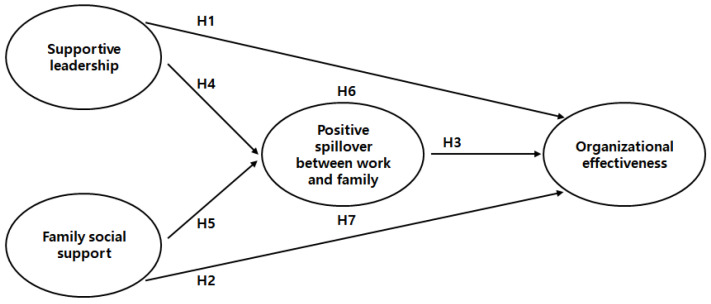
The study’s proposed conceptual framework.

**Figure 2 behavsci-13-00639-f002:**
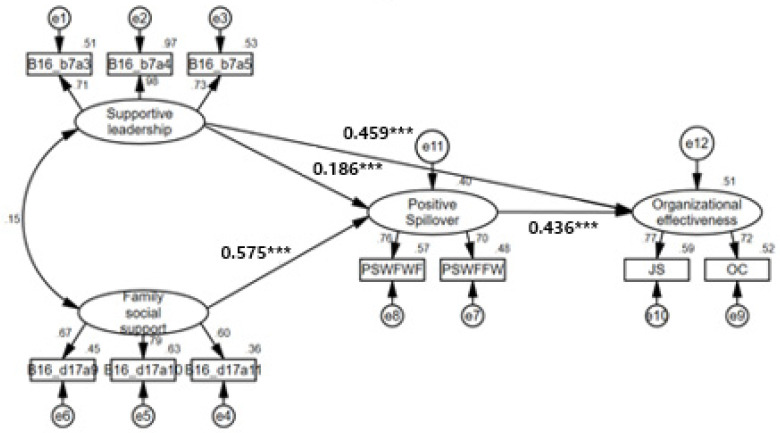
The results of the modified structure model. Standardized coefficients, *** *p* < 0.001.

**Table 1 behavsci-13-00639-t001:** Item Components of Measures.

Variables	Items	Cronbach’s Alpha	McDonald’s ω
Supportive leadership	1. My boss allows me to change my personal work hours, overtime or vacation plans. 2. My boss gives me a good deal of both work and family life. 3. My boss listen to my problems well	0.84	0.85
Family social support	1. Family members either replace or help me with the roles I should play in the home. 2. Family members understand me even if I can’t participate in family events due to work day. 3. I can work late without worrying about housework.	0.71	0.71
Job satisfaction	1. coworkers 2. wages 3. supervisors 4. duties 5. job environment 6. working hours	0.82	0.82
Organizational commitment	1. I feel a strong sense of belonging in my organization. 2. My organization seems like a part of my family. 3. I am proud of my organization. 4. I view my organization’s problems as my own problems. 5. This organization is personally meaningful for me. 6. I would be very happy if I could build the remainder of my career in this organization.	0.90	0.90
Positive spillover work to family	1. The things you do at work give your life value and vitality. 2. What you do at work helps you to be a more interesting person at home. 3. Having a nice day at the workplace helps you to be a better companion at home.	0.83	0.84
Positive spillover family to work	1. The responsibility to support family makes your work harder in the workplace. 2. Recognition of the value of your work from your family members makes your work harder in the workplace. 3. When you have a problem at work, members of your family always talk and advise you about it.	0.58	0.66

**Table 2 behavsci-13-00639-t002:** Correlation.

Variables	1	2	3	4	5	6
1. Supportive Leadership	1					
2. Family social support	0.126 **	1				
3. Positive spillover work to family	0.222 **	0.341 **	1			
4. Positive spillover family to work	0.138 **	0.435 **	0.521 **	1		
5. Job satisfaction	0.479 **	0.168 **	0.328 **	0.174 **	1	
6. Organizational commitment	0.352 **	0.225 **	0.450 **	0.270 **	0.553 **	1

** *p* < 0.01.

**Table 3 behavsci-13-00639-t003:** Analysis of the model fit index for the research model and modified model.

Model	χ^2^(p)	χ^2^/df(Q Value)	GFI	TLI	CFI	SRMR	RMSEA
Research Model	177.556 (0.000)	6.123	0.965	0.932	0.956	0.046	0.073
Modified Model	179.703 (0.000)	5.990	0.964	0.934	0.956	0.046	0.072

**Table 4 behavsci-13-00639-t004:** The direct, indirect, and total effects between the latent variables.

Hypothesis	Path	Path Coefficient		
		Direct Effect	Indirect Effect	Total Effect
6	SL → PSWF → OE	0.459 **	0.081 **	0.540 **
7	FSS → PSWF → OE	-	0.250 **	0.250 **

** *p* < 0.01. Note. SL: supportive leadership, FSS: family social support, PSWF: positive spillover between work and family, OE: organizational effectiveness.

**Table 5 behavsci-13-00639-t005:** The mediating effect of PSWF using bootstrapping.

Hypothesis	Path	Product of Coefficient			BC 99% CI	
		ab	S.E.	*p*	Lower	Upper
6	SL → PSWF → OE	0.081	0.039	0.000	0.434	0.634
7	FSS → PSWF → OE	0.250	0.035	0.000	0.167	0.346

Note. ab = Completely standardized estimate, S.E. = Standard Errors, BC = Bias-corrected, CI = Confidence Interval, SL: supportive leadership, FSS: family social support, PSWF: positive spillover between work and family, OE: organizational effectiveness.

## Data Availability

The data presented in this study are available from Korean Women Manager Panel (https://gsis.kwdi.re.kr/kwmp/about/kwmpIntro.do (accessed on 19 June 2023)).

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
