# Peer review of "The Influences of Supportive Leadership and Family Social Support on Female Managers’ Organizational Effectiveness: The Mediating Effect of Positive Spillover between Work and Family"

_behavsci, 2023, doi:10.3390/bs13080639_

Round 1
Reviewer 1 Report
Dear authors,
First of all, congratulations on your work. I believe it is a rigorous and comprehensive approach to understanding different support models in the organizational field. I think it is a well-argued and well-founded piece in its introduction, and it presents coherent hypotheses.
I also find it to be solid in its methodological structure, although I would like to offer some constructive feedback on it.
1. Regarding the EFA, you are working with Likert-5 items. This type of items results in ordinal variables, making exploratory analysis more appropriate using a polychoric correlation matrix and extraction methods designed for categorical variables, such as ULS or DWLS. This type of analysis can be easily conducted using open-source software like Factor Analysis (https://psico.fcep.urv.cat/utilitats/factor/Download.html), and there are recent studies that apply these methods: https://www.psicothema.com/pi?pii=4733.
2. Building upon the previous comment, it would be appropriate to also provide McDonald's Omega as a reliability index, in addition to Cronbach's alpha. Factor or JASP (https://jasp-stats.org/) are software tools that allow easy access to these values. As previously mentioned, both are open-source software.
3. In the data analysis, it would be helpful to indicate the cutoff criteria used for interpreting the fit indices of the measurement and SEM models.
4. In this regard, the TLI index, although robust, is somewhat low in the SEM models (below .95). It would be advisable to acknowledge this fact and provide a justification.
5. It would be useful to report the explained variance of the endogenous variables in the SEM model using the R^2 index to assess the model's explanatory power.
6. Finally, it would be beneficial to strengthen the discussion section. Your work is complex and in-depth, with a substantial number of hypotheses, and I believe a more thorough examination in this section would be helpful.
I reiterate my positive appraisal of the presented work and hope that these comments prove useful.
Best regards.
Author Response
Point 1: Regarding the EFA, you are working with Likert-5 items. This type of items results in ordinal variables, making exploratory analysis more appropriate using a polychoriccorrelation matrix and extraction methods designed for categorical variables, such as ULS or DWLS. This type of analysis can be easily conducted using open-source software like Factor Analysis (https://psico.fcep.urv.cat/utilitats/factor/Download.html), and there are recent studies that apply these methods: https://www.psicothema.com/pi?pii=4733.
Response 1: Thank you very much for your kind and constructive comments on more rigorous data analysis in this study. As you suggested, we additionally confirmed the result of EFA using FACTOR and provided the results in 4.2 Exploratory Factor Analysis on page 13.
Point 2: Building upon the previous comment, it would be appropriate to also provide McDonald's Omega as a reliability index, in addition to Cronbach's alpha. Factor or JASP (https://jasp-stats.org/) are software tools that allow easy access to these values. As previously mentioned, both are open-source software.
Response 2: We provided McDonald's ω as a reliability index in the table 1 on page 11.
Point 3: In the data analysis, it would be helpful to indicate the cutoff criteria used for interpreting the fit indices of the measurement and SEM models.
Response 3: We indicated the cutoff criteria for the fit indices of the measurement and SEM models. The acceptable threshold levels of fit indices are described in 3.3 data analysis part on page 12.
Point 4: In this regard, the TLI index, although robust, is somewhat low in the SEM models (below .95). It would be advisable to acknowledge this fact and provide a justification.
Response 4: The fit indices are added in 3.3 data analysis part on page 12.
Point 5: It would be useful to report the explained variance of the endogenous variables in the SEM model using the R^2 index to assess the model's explanatory power.
Response 5: We added explanation for explanatory power of endogenous variable (R2) in 4.5 Hypotheses Testing part on page 15.
Point 6: Finally, it would be beneficial to strengthen the discussion section. Your work is complex and in-depth, with a substantial number of hypotheses, and I believe a more thorough examination in this section would be helpful.
Response 6: We revised the discussion section with more recent literature on pages 16-17.
* We marked blue on the parts that were revised.
Reviewer 2 Report
This empirical paper proposes to examine the effect of supportive leadership and family social support on female managers’ organizational effectiveness using one time design, and data from 974 married female 436 managers out of 1,384 people who continued to maintain their jobs. Authors propose a very interesting study to understand the mechanism explaining the role of PSWF (mediation) in the relationship between supportive leadership and organizational effectiveness, on one hand, and between family social support and organizational effectiveness on the other hand. Here is some advice on how to improve your paper especially conceptual, methodological and statistical aspects which is very important to set the direction for the rest of the paper:
1. The multitude of theories used in the paper makes it difficult to understand the main line of reasoning and construction of the hypotheses.
2. It is very remarkable the low score of the reliability coefficient of positive spillover family to work scale (.58). It seems that the translation of the scales generated disparity in the meaning of the items’ formulation. Did the authors follow a rigorous translation procedure measurement scale?
3. The correlation table reveals moderate correlation between organizational commitment and job satisfaction; How this result can be interpreted?
4. As noted in the limitation section of the paper, measurement of performance by the employee himself and not by his supervisor is a major limitation. In addition, performance measurement does not relate to objective elements, it’s just a perception of one's own performance.
5. It is not appropriate to combine two theoretically distinct concepts such as organizational commitment and job satisfaction.
6. Confirmatory factor analysis (CFA) is not appropriate. It is important to compare the null model, the four-factor model (supportive leadership, family social support, positive spillover, organizational effectiveness) and the six-factor model (supportive leadership, family social support, positive spillover work to family, positive spillover family to work, organizational commitment, job satisfaction).
7. Discuss the results in light of previous studies and add some recent references.
Author Response
Point 1: The multitude of theories used in the paper makes it difficult to understand the main line of reasoning and construction of the hypotheses.
Response 1: We revised the 2.4 Positive Spillover between Work and Family part (p. 6), which caused confusion due to a number of theories, to make the main reasoning and construction of the hypothesis clearer.
Point 2: It is very remarkable the low score of the reliability coefficient of positive spillover family to work scale (.58). It seems that the translation of the scales generated disparity in the meaning of the items’ formulation. Did the authors follow a rigorous translation procedure measurement scale?
Response 2: We provided the criteria for Cronbach’s alpha and the McDonald's ω in the 3.2 Measure part on page 10.
In this study, we did not translate the scales because we used the panel survey data conducted to Koreans in Korea.
Point 3: The correlation table reveals moderate correlation between organizational commitment and job satisfaction; How this result can be interpreted?
Response 3: We provided relevant references in the part of 4.1. Descriptive Statistics, Normality, Correlations, and Multicollinearity on page 12. In addition, given the moderate correlation (.553) between organizational commitment and job satisfaction examined, the presence of multicollinearity was evaluated. No multicollinearity was found, with an acceptable range of .653~.948 for the tolerance value and an acceptable range of 1.055~1.532. for the VIF value of all the variables.
Point 4: As noted in the limitation section of the paper, measurement of performance by the employee himself and not by his supervisor is a major limitation. In addition, performance measurement does not relate to objective elements, it’s just a perception of one's own performance.
Response 4: In this study, we used the panel survey data measured by employees’ perceptions. In this respect, we described this issue in the directions for future research as follows on page 20. Using various job performance indicators in different levels will lead to a more systematic understanding of the impact of supportive leadership, family social support, and positive spillover between work and family on organizational effectiveness.
Point 5: It is not appropriate to combine two theoretically distinct concepts such as organizational commitment and job satisfaction.
Response 5: We revised this part on page 3 by supplementing more references showing that organizational effectiveness consists of job satisfaction and organizational commitment.
Point 6: Confirmatory factor analysis (CFA) is not appropriate. It is important to compare the null model, the four-factor model (supportive leadership, family social support, positive spillover, organizational effectiveness) and the six-factor model (supportive leadership, family social support, positive spillover work to family, positive spillover family to work, organizational commitment, job satisfaction).
Response 6: The PSWF (positive spillover between work and family) and work effectiveness are second-order factor models that consist of second-order latent variables. In this study, PSWF and organizational effectiveness both had two first-order factors and one second-order factor; the first-order factors were positive spillover from work to family and positive spillover from family to work for the former, and organizational commitment and job satisfaction for the latter. After conducting a second-order confirmatory factor analysis, we performed a first-order confirmatory factor analysis.
Point 7: Discuss the results in light of previous studies and add some recent references.
Response 7: Based on the results of previous studies, the discussion part was revised, and recent references were added on pages 16-17.
* We marked blue on the parts that were revised.
Round 2
Reviewer 2 Report
The authors responded to my requests for improvement of the paper. I have no additional requests.